# Human Norovirus: Experimental Models of Infection

**DOI:** 10.3390/v11020151

**Published:** 2019-02-12

**Authors:** Kyle V. Todd, Ralph A. Tripp

**Affiliations:** Department of Infectious Diseases, College of Veterinary Medicine, University of Georgia, Athens, GA 30602, USA; ktodd@uga.edu

**Keywords:** norovirus, human norovirus, animal models, reverse genetics, vaccine development

## Abstract

Human noroviruses (HuNoVs) are a leading cause of acute gastroenteritis worldwide. HuNoV infections lead to substantial societal and economic burdens. There are currently no licensed vaccines or therapeutics for the prevention or treatment of HuNoVs. A lack of well-characterized *in vitro* and *in vivo* infection models has limited the development of HuNoV countermeasures. Experimental infection of human volunteers and the use of related viruses such as murine NoV have provided helpful insights into HuNoV biology and vaccine and therapeutic development. There remains a need for robust animal models and reverse genetic systems to further HuNoV research. This review summarizes available HuNoV animal models and reverse genetic systems, while providing insight into their usefulness for vaccine and therapeutic development.

## 1. Introduction

Human noroviruses (HuNoVs) are non-enveloped, single-stranded, positive-sense, RNA viruses belonging to the *Caliciviridae* family [1,2,3]. Their 7.5–7.7 kb genomes contain three open reading frames (ORFs) (Figure 1a) [4]. ORF1 codes for the six nonstructural proteins, in order from N-terminus to C-terminus: p48, NTPase, p22, VPg, 3C-like protease (3CL^pro^), and RNA dependent RNA polymerase (RdRp) [5]. Subgenomic RNA, containing ORFs 2 and 3, codes for the major and minor structural proteins, VP1 and VP2 (Figure 1a) [6]. The *Norovirus* (NoV) genus is divided into seven genogroups (GI–GVII) based on VP1 amino acid homology [7,8,9]. Each genogroup is made up of genotypes GI (n = 9), GII (n = 25), GIII (n = 2), GIV (n = 2), GV (n = 1), GVI (n = 2), and GVII (n = 1), which contain individual virus strains [8,10]. GI, GII, and to a lesser extent GIV NoVs cause disease in humans. GI NoVs only infect humans; however, GII and GIV contain NoVs that infect cats, dogs, pigs, and humans [8]. NoVs are understood to be species-specific [8,11]. Interestingly, GII HuNoVs bind to porcine gastric mucins [12,13,14] and can infect pigs, but robust zoonotic and reverse zoonotic transmissions have not been reported.

HuNoVs are the leading cause of acute gastroenteritis worldwide [15,16,17,18,19]. HuNoVs transmit through the fecal–oral route upon ingestion of the encapsidated virions. Following a 24–48 h incubation period, HuNoVs cause symptomatic diarrhea and vomiting for the next 12–60 h [20,21,22]. The infection is self-limiting within a few days, but the virus continues to be shed in the feces for the next few weeks in immunocompetent patients [23,24,25,26]. Annually, there are approximately 700 million infections that result in >200,000 deaths and have an economic burden of >$64 billion [27]. Humans of all age groups are susceptible to HuNoV infection, but children, the immunocompromised, and the elderly are more likely to develop severe disease and therefore are groups of interest for vaccination. Currently, there are no licensed vaccines or therapeutics for the prevention or treatment of HuNoV. Nearly all candidate HuNoV vaccines are subunit vaccines generated from virus-like particle (VLP) constructs. HuNoV VLPs assemble spontaneously after the expression of either VP1, or VP1 and VP2. The immunogenicity of HuNoV VLPs in BALB/c mice upon oral [28,29], intradermal [29,30], intramuscular [31], intranasal [31,32,33,34], and sublingual [32] administration have been studied. The immune responses following intranasal administration of HuNoV VLPs to guinea pigs have also been evaluated [34,35]. VLP vaccination using mucosal adjuvants in gnotobiotic (Gn), germ-free, piglets was evaluated after oral vaccination followed by two intranasal boost immunizations [36]. Following homologous GII.4 challenge, no VLP + adjuvant immunized piglets shed virus and only one had diarrhea [36]. The immunogenicity and protective efficacy of oral, intranasal, and intramuscular bovine NoV VLP immunizations were tested in Gn calves, providing partial protection from disease after homologous bovine NoV challenge [37]. Intramuscular vaccination of chimpanzees with GI.1 VLPs, but not GII.4 VLPs, protected them from homologous HuNoV challenge [38]. Despite the studies completed, no standard animal models have been established for testing HuNoV candidate vaccines. There are currently two VLP vaccines in clinical trials. One is an aluminum hydroxide adjuvanted bivalent GI.1 and GII.4 VLP vaccine [15,39,40]. It has been shown to be immunogenic in rabbits by intranasal and intramuscular administration routes [41]. The other is an adenovirus-vectored GI.1 VP1 VLP vaccine which completed phase I clinical trials [42], but was not tested in any animal models prior to experimental human immunizations. While HuNoV VLPs stimulate antibody responses in BALB/c mice and rabbits, it has not been reported whether or not these animals are susceptible to HuNoV challenge [43,44]. Thus, the implications of vaccination and protection in these studies are unclear. Animal models used in HuNoV studies are summarized in Table 1. Translatable animal models may be useful for evaluating candidate HuNoV vaccines and therapeutics.

HuNoV vaccine efforts are further impeded by an inability to grow HuNoV in a vaccine-approved cell line. This prevents the production of whole virus vaccines (inactivated or live-attenuated). Currently, B cell [45,46,47] and human intestinal enteroid [48,49,50,51] models are used for propagating HuNoV *in vitro*; however, these systems are not optimal for HuNoV vaccine production [52]. Work to develop a reverse genetics system in lieu of an *in vitro* cell culture system is in progress to bypass the requirement for natural infection. A reverse genetics system provides a platform for studying the role of individual HuNoV proteins in the context of a HuNoV infection, which may inform the generation of attenuated HuNoV vaccines.

## 2. Large Animal Models

### 2.1. Non-Human Primates (NHPs)

Initial efforts to develop large animal models began with unsuccessful attempts to infect non-human primates (NHPs), e.g., rhesus monkeys and baboons, with HuNoV [43]. It was then discovered that chimpanzees produce serum antibodies and shed virus upon oral HuNoV infection, but do not develop gastroenteritis [43]. A single passage of HuNoV in chimpanzees did not alter virus shedding, symptom presentation, or antibody production, indicating a lack of adaptation to the chimpanzee host [43]. Intravenous (i.v.) administration of HuNoV resulted in asymptomatic infection and antibody responses that were dominated by serum IgM and IgG [38]. HuNoV was shed in the feces at similar times following oral or i.v. infection, but the duration of shedding was increased in i.v. infected chimpanzees. Viremia was not detected at any timepoints post-infection (pi), although viral RNAs were detectable in liver tissue. Interestingly, HuNoV antigens were detected in the duodenum, jejunum, and lamina propria (dendritic cells) despite no histological changes to the intestines [38]. Lastly, an infectious dose 50% (ID_50_) of 4.0 × 10^7^ genome equivalents (g.e.) was established for i.v. infection of chimpanzees [38]. A lack of disease presentation greatly limits the utility of the chimpanzee model for HuNoV studies.

Adult and neonatal pigtail macaques are also susceptible to HuNoV infection [53]. The onset of virus shedding and its duration, as well as the presence and duration of diarrhea are similar between adults and neonates [53]. Curiously, one adult pigtail macaque vomited during the study, a symptom not previously observed in any other HuNoV animal model [53]. Further evaluation of the pigtail macaque animal model may prove fruitful if these animals consistently display symptomatic diarrhea and vomiting as a result of HuNoV infection. Experimental oral infection of other NHPs (common marmosets, cotton-top tamarins, and cynomolgus macaques) did not induce the production of HuNoV-specific antibodies or robust virus shedding [54]. However, a single rhesus macaque from this study shed virus for over two weeks and maintained robust IgG titers until the end of the experimental timecourse [54].

### 2.2. Gnotobiotic (Gn) Piglets

Gn piglets develop diarrhea, shed virus, and have detectable levels of HuNoV in the intestines upon oral infection [55,56,57,58,59,60,61]. Although low, HuNoV-specific serum and mucosal antibodies have been reported in Gn piglets [61]. Additionally, HuNoV remains infectious after two passages in Gn piglets [55]. Like chimpanzees, the passage of HuNoV in Gn piglets did not improve any infection parameters. An ID_50_ for oral HuNoV infection of Gn piglets was established at <2.74 × 10^3^ g.e. for 4–5 day old piglets and 6.43 × 10^4^ g.e. for 33–34 day old piglets [57]. Experimentally, an ID_50_ of 18 virions was established for GI.1 Norwalk virus [62], while another group determined the Norwalk virus ID_50_ to range between 1.32 × 10^3^–2.80 × 10^3^ viral particles [24]. To date, no ID_50_ has been reported for any GII.4 HuNoV. The Gn piglet model has been used to test adjuvanted HuNoV VLPs [36] and the inactivation of HuNoV by high pressure processing [63]. Work with the Gn piglet model has also been expanded to include *RAG2*/*IL2RG* double knockout (KO) piglets that experience prolonged HuNoV antigen retention in the intestines and asymptomatic virus shedding [58]. In contrast, piglet models with natural flora, such as the miniature piglet model, lack substantial disease presentation [64]. Simvastatin, a cholesterol-reducing statin, has been shown to increase HuNoV infectivity and disease in the Gn piglet model [57,65]. These results strengthen a report that statin use is a risk factor for enhanced HuNoV disease in humans [66]. Simvastatin has immunosuppressive properties and has been linked to the downregulation of interferon-α (IFNα) and major histocompatibility complex II expression [57,65]. Oral administration of IFNα augmented the immune responses of HuNoV-infected Gn piglets, decreasing the onset of virus shedding [65]. The Gn piglet model has also been used to study how individual commensal bacteria affect HuNoV replication *in vivo*. The histo-blood group antigen (HBGA)-expressing *Enterobacter cloacae* inhibited HuNoV replication and virus shedding, possibly because HuNoV binds to HBGAs on the bacteria rather than their target eukaryotic cells [67]. Interestingly, this result contrasts with an *in vitro* study that demonstrated the enhancement of HuNoV infection by *E. cloacae* [46]. The effects of gastrointestinal bacteria (both commensal and pathogenic) on enteric viral infections represent important considerations relevant for HuNoV animal model development [68,69,70]. Additionally, the role of HBGAs as attachment factors influencing susceptibility for HuNoV infections is important and has been reviewed [71,72]. HuNoV infection of HBGA-typed Gn piglets has been studied [73], but the implications of HBGA expression for HuNoV infection have not been previously studied in the other animal models.

### 2.3. Gn Calves

Gn calves develop diarrhea and shed HuNoV for up to 6 days after oral infection [74]. Additionally, HuNoV antigen is readily detectable in enterocytes and the lamina propria of infected calves [74]. Notably, intestinal damage and intestinal/serum IgA and IgG production are observed in this animal model [74].

## 3. Small Animal Models

Attempts to infect adult and suckling mice, kittens, guinea pigs, or rabbits with HuNoV have been unsuccessful [43]. To date, the only small animal model described is a recombination activation gene (*Rag*^−/−^) and common gamma chain (*γc*^−/−^) deficient BALB/c mouse [44]. The double KO supported a mixed culture of GII HuNoVs through intraperitoneal (i.p.), but not oral infection. Interestingly, dual administration (oral and i.p.) increased viral loads above those observed after i.p. injection alone [44]. The *Rag*^−/−^*γc*^−/−^ BALB/c mouse model has been implemented to demonstrate the antiviral properties of the nucleoside analogue, 2′-C-methylcytidine (2CMC), *in vivo* [47]. Commercially available wildtype BALB/c and *Rag*^−/−^*γc*^−/−^ C57BL/6J × C57BL/10SgSnAi mice do not support HuNoV, suggesting the requirement for a compromised immune system and possibly other unresolved host factors that may permit virus infection and replication [44].

## 4. Reverse Genetic Systems

It has been shown that HuNoV RNA isolated from stool can produce all of the structural and nonstructural proteins upon transfection into either Huh7 or Caco-2 cells [75]. However, only a single cycle of replication occurs due to blocks at the receptor binding or uncoating stages [75]. Although inefficient, 100–150 positive cells could be detected after transfecting 5.0 × 10^8^ HuNoV RNA molecules into 10^5^ cells [75]. This observation initiated the development of HuNoV reverse genetics. The first report of a HuNoV replicon system utilized a recombinant Vaccinia virus strain, modified Vaccinia Ankara (MVA), to drive a T7 promoter-controlled plasmid containing a GI.1 Norwalk virus full-length clone in HEK293T cells (Figure 1b) [76]. This system produced approximately 8.5 × 10^4^ HuNoV particles from 12 T-75 flasks [76]. The first GII HuNoV infectious clone was created shortly after using a similar Vaccinia virus T7 promoter system (vTF7) with the GII.3 U201 strain in HEK293T cells (Figure 1b) [77]. However, the particles formed using this system were found to be less dense than naturally isolated HuNoV (1.32 g/cm^3^ versus 1.39–1.40 g/cm^3^). With both methods, Vaccinia caused HuNoV-independent cytopathic effects at 72 hpi, limiting the utility of these systems.

Subsequently, a helper virus-free GI.1 Norwalk virus system was developed (Figure 1c) [78]. In vitro transcription of a plasmid using Vaccinia virus-T7 produced a recombinant RNA that contained the full-length genome with a neomycin resistance gene (*neo*) inserted into the *VP1* gene. Transfection of this recombinant RNA produced the nonstructural proteins and VP2. Disruption of the *VP1* gene prevents the use of this system to isolate infectious virus, because the major capsid protein is compromised. This system was successfully used to transfect Huh7 and BHK21 cells, but not Vero, 293, 293T, or LLC-PK cells [78]. The transfected cells remained stably transfected after at least 100 passages, with levels of 2.6 × 10^11^ g.e./1 µg of RNA and 8.0 × 10^8^ g.e./1 µg of RNA produced in Huh7 and BHK21 cells, respectively. The implementation of this system was used to determine that Norwalk virus was sensitive to 72 h treatment with IFNα (effective dose 50% (ED_50_) levels of 2 units/mL and 20 units/mL for Huh7 and BHK21 cells, respectively) [78]. Another study using this system demonstrated that Norwalk virus produced in Huh7 cells was also sensitive to IFNγ, ribavirin, and mycophenolic acid [79]. Nucleoside analogue inhibition of viral transcription by 2CMC [80] or 7-deaza-2′-C-methyladenosine [81] treatment decreased Norwalk virus production in this replicon system. Additionally, a GI.1 Norwalk virus replicon-expressing human gastric tumor-1 system was adapted from the Huh7 system to explore the development of resistance to rupintrivir, a viral protease inhibitor [82].

The first true helper virus-free system used the mammalian EF-1α promoter to drive HuNoV RNA production (Figure 1d) [16]. This system also supported infectious murine NoV (MuNoV) production [16]. Originally developed for the GII.3 U201 strain, the production of full-length genomes per 10^6^ cells amounted to 8.0 × 10^4^, 1.4 × 10^4^, 2.4 × 10^2^, and 1.3 × 10^1^ copies in COS7, 293T, Huh7, and Caco-2 cells, respectively. As evidenced by MuNoV production using this system, HuNoV virion production may be up to 10-fold less than the corresponding RNA levels [16]. Differences in virus production levels can be explained by the transformed cell lines (COS7 and HEK293T) expressing the SV40 T antigen, which helps drive the SV40 promoter in the expression system [16]. The utility of this system was enhanced by adding a *GFP* gene between the *NTPase* and *p22* viral genes, creating a GFP reporter infectious clone [16]. However, the insertion of the *GFP* gene resulted in up to 50-fold fewer HuNoV virions being produced [16]. Despite an inability to detect the capsid proteins by Western blot or immunofluorescence assays, an ORF2 GFP reporter plasmid indicated that the structural proteins were being produced [16]. Unfortunately, attempts to adapt the system to GI.1 NV68, GII.P4-GII.3 chimera TCH04-577, and GII.4 Saga1 viruses resulted in 10- to 1000-fold less HuNoV production [16]. Although the HuNoV RNA from these recombinant virions is infectious, upon transfection, the infectivity of the intact virions has not been tested [16].

The clinical relevancy of HuNoV reverse genetic systems was an issue, until the development of a GII.4 Sydney 2012 infectious clone (Figure 1e) [83]. Structural proteins were readily detected upon transfection of the plasmid into Caco-2 cells [83]. The insertion of a *GFP* sequence between the *NTPase* and *p22* viral genes generated a fluorescent reporter system. However, this negatively affected viral structural protein synthesis through an unknown mechanism, as previously observed [16,83]. The utility and robustness of this system remain unclear as studies quantifying virus production and demonstrating the passage of the recombinant virions still need to be completed.

## 5. Conclusions

The development of animal models and reverse genetics for HuNoV are significant milestones that are needed to drive HuNoV pathogenesis studies and vaccine development. Currently, the pigtail macaque is the most promising NHP infection model for HuNoV. Expanded studies to test levels of virus replication, the detection of HuNoV in tissues, the presence of antibodies at mucosal surfaces, and infection with clinically relevant GII.4 HuNoVs are needed. Vomiting as a result of HuNoV infection is a unique observation for this animal model. Epidemiologically, one criterion for the determination of HuNoV as the etiological agent causing an outbreak is that ≥50% of affected persons present with vomiting [21,22]. The precise cause of HuNoV-induced vomiting is unclear and has not been as readily investigated as HuNoV-associated diarrhea. Of the models described, only NHPs and piglets have an emetic response, while calves and mice are understood to have no or a greatly limited emetic response. This precludes HuNoV-associated vomiting from being studied in calves or mice. Even so, both the Gn piglet and the under-studied Gn calf model are encouraging large animal models for HuNoV infection. Prolonged virus shedding and diarrhea in response to infection and the production of serum and mucosal antibodies underscore the usefulness of these models. Further development of the pigtail macaque, Gn piglet, and Gn calf models may yield robust models of infection that can bolster experimental human infection studies. A drawback to the current large animal models for HuNoV infection is their accessibility. NHPs are costly and studies involving such models raise ethical concerns, potentially limiting their utility. Likewise, Gn animals require specific handling and facilities in order to maintain bacteria-free animals [84,85]. Lack of commensal bacteria in these animals impacts HuNoV infection [64,67] and likely impairs mucosal immune responses especially in the gastrointestinal tract. However, Gn models provide the opportunity to study HuNoV infections in the context of controlled bacterial populations. Gn animals can be colonized with single bacterial species or complex bacterial populations by fecal transplants, from similar or even dissimilar mammalian hosts. Therefore, the reconstitution of Gn piglets or calves with a human microbiota favorable for HuNoV infection may enhance these models. The benefits of large animal models for HuNoV infection include more extensive tissue sampling and ease of test subject procurement, as compared to human experimental infection studies. Further advantages to accessibility and cost are realized when working with a small animal model such as the *Rag*^−/−^*γc*^−/−^ BALB/c mouse. Unfortunately, transmission studies cannot be completed in these mice because they cannot be infected orally. Also, *Rag*^−/−^*γc*^−/−^ mice lack the ability to produce numerous cytokines and mature B and T cells. Immune responses to infection are therefore not representative of typical HuNoV infection. Expansion of studies with the large animal models (pigtail macaque, Gn piglet, and Gn calf) and the development of small animal models beyond the *Rag*^−/−^*γc*^−/−^ BALB/c mouse will provide opportunities to standardize the preclinical HuNoV therapeutic and vaccine pipelines.

HuNoV reverse genetic systems provide a platform for *in vitro* infection in lieu of a classical infection model. However, the adoption of all current HuNoV reverse genetic systems is impeded by their inefficiency compared to the recovery of virus from stool or MuNoV reverse genetic systems [86,87,88]. Additionally, lack of a recognized receptor for HuNoV infection and other gaps in the understanding of HuNoV biology have limited the development of both reverse genetic systems and cell culture models. The discovery of CD300lf as a proteinaceous receptor for MuNoV [89,90] has greatly advanced the study of MuNoV. This type of breakthrough for HuNoV is poised to revolutionize the development of HuNoV *in vitro* propagation systems. Further studies are needed to optimize the HuNoV reverse genetic systems, which may provide a platform for developing live-attenuated HuNoV vaccines, an untapped and under-investigated area. MuNoV work in this sector has previously identified sites for genetic engineering that may prove useful for HuNoV vaccine studies [91,92].

## Figures and Tables

**Figure 1 viruses-11-00151-f001:**
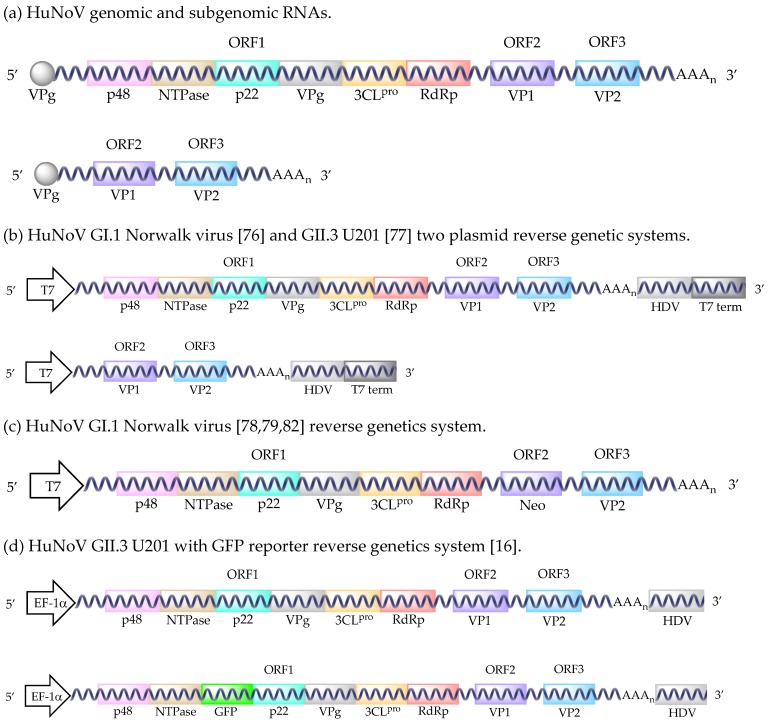
HuNoV reverse genetic systems. HuNoV genomic and subgenomic RNAs (**a**). HuNoV GI.1 Norwalk virus and GII.3 U201 two plasmid reverse genetic systems (**b**). HuNoV GI.1 Norwalk virus reverse genetics system (**c**). HuNoV GII.3 U201 with GFP reporter reverse genetics system (**d**). HuNoV GII.4 Sydney with GFP reporter reverse genetics system (**e**). Abbreviations: T7 polymerase promoter sequence (T7), hepatitis delta virus ribozyme (HDV), T7 terminator sequence (T7 term), human elongation factor-1 alpha promoter sequence (EF-1α), cytomegalovirus promoter sequence (CMV), bovine growth hormone polyadenylation signal (BGH pA).

**Table 1 viruses-11-00151-t001:** Animal models of human norovirus (HuNoV) infection and disease. Non-human primates (NHP), gnotobiotic (Gn) piglet, natural flora piglet, Gn calf, and mouse animal models are grouped by shading. Abbreviations: humanized (Hum.), intravenous (i.v.), intraperitoneal (i.p.), days post-infection (dpi), not applicable (N/A), not reported (NR), data not shown (DNS), respectively (resp.), abnormal histopathology (AH), antibodies (Abs).

Model	Ref.	HuNoV Genotype (Strain)	Infect. Route	Fecal Virus Shedding	Shedding Start	Shedding Duration	Diarrhea	Day 1st Detected	Diarrhea Duration	Vomit	AH	HuNoV in Tissue	HuNoV in Serum	Serum Abs	Mucosal Abs
Chimpanzees	[38]	GI.1 (Norwalk virus)	i.v.	Yes; qRT-PCR	2, 3, 3, 3, 4, or 5 dpi	22, 23, 31, 42, 17, or 22 days resp.	No	N/A	N/A	No	No	Yes	NR	IgM and IgG	NR
Chimpanzees	[43]	GI.1 (Norwalk virus) passage 0	oral	Yes; radioimmunoassay	2, 2.5, or 3 dpi	2.5, 1.5, or 1 day(s) resp.	No	N/A	N/A	No	NR	NR	NR	IgG	NR
Chimpanzees	[43]	GI.1 (Norwalk virus) passage 1	oral	Yes; radioimmunoassay	3 or 4 dpi	3 or 2 days resp.	No	N/A	N/A	No	NR	NR	NR	IgG	NR
1–3 month old pigtail macaques	[53]	GII.3 (Toronto)	oral	Yes; RT-PCR, ELISA, and EM	1, 2, or 21 dpi	21, 13, or 1 day(s) resp.	Yes	2 dpi	1 or 2 day(s)	No	NR	NR	NR	NR	NR
Pigtail macaques (adult)	[53]	GII.3 (Toronto)	oral	Yes; RT-PCR and ELISA	1 dpi	21 days	Yes	3 dpi	1 day	Yes	NR	NR	NR	IgG	NR
Common marmosets	[54]	GI.1 (Norwalk virus)	oral	Yes; RT-PCR	1 dpi	3 days	No	N/A	N/A	NR	NR	NR	NR	No	No
Cotton-top tamarins	[54]	GI.1 (Norwalk virus)	oral	Yes; RT-PCR	1 dpi	4 days	No	N/A	N/A	NR	NR	NR	NR	No	No
Cynomolgus macaques	[54]	GI.1 (Norwalk virus) or GII.4 (Grimsby)	oral	No	N/A	N/A	No	N/A	N/A	NR	NR	NR	NR	No	No
Rhesus macaques	[54]	GI.1 (Norwalk virus)	oral	Yes; RT-PCR	1 dpi	18 days	No	N/A	N/A	NR	NR	NR	NR	IgM and IgG	No
33 day old Gn piglets	[36]	GII.4 (HS66)	oral	Yes; RT-PCR	NR	NR	Yes	DNS	2 days	NR	NR	NR	NR	No	IgM
Gn piglets	[55]	GII.4 (HS66) passage 0	oral	Yes; RT-PCR	1 dpi	3 days	Yes	2 dpi	2 days	NR	Yes	Yes	Yes	IgG	NR
Gn piglets	[55]	GII.4 (HS66) passage 1	oral	Yes; RT-PCR	1 dpi	2 days	Yes	2 dpi	3 days	NR	NR	Yes	No	IgG	NR
Gn piglets	[55]	GII.4 (HS66) passage 2	oral	Yes; RT-PCR	2 dpi	2 days	Yes	2 dpi	1 day	NR	NR	Yes	No	IgG	NR
Gn piglet	[55]	GII.4 (HS66) passage 0	i.v.	Yes; RT-PCR	DNS	DNS	Yes	DNS	DNS	NR	NR	NR	NR	NR	NR
Gn piglet	[55]	GII.4 (HS66) passage 1	i.v.	No	N/A	N/A	Yes	DNS	DNS	NR	NR	NR	NR	NR	NR
6 day old Gn piglets	[56]	GII.12 (HS206)	oral	Yes; qRT-PCR	1, 2, 3, or 3 dpi	3, 3, 8, or 14 days resp.	Yes	1 dpi	3 days	NR	NR	NR	NR	NR	NR
4–5 day old Gn piglets	[57]	GII.4 (092895; 2006b variant)	oral	Yes; qRT-PCR	2 dpi	DNS	Yes	4 dpi	DNS	NR	Yes	NR	NR	NR	NR
33–34 day old Gn piglets	[57]	GII.4 (092895; 2006b variant)	oral	Yes; qRT-PCR	3 dpi	DNS	Yes	DNS	DNS	NR	Yes	NR	NR	NR	NR
6–7 day old Gn piglets	[58]	GII.4 (092895; 2006b variant)	oral	Yes; qRT-PCR	1 dpi	max = 16 days	Yes	DNS	DNS	NR	NR	Yes	Yes	NR	NR
28 day old Gn piglets	[59]	GII.4 (KU131206)	oral	Yes; qRT-PCR	1 or 2 dpi	1, 2, or 3 days	Yes	2 or 3 dpi	1 or 2 day(s)	NR	Yes	Yes	Yes	NR	NR
32–33 day old Gn piglets	[60]	GII.4 (092895; 2006b variant)	oral	Yes; qRT-PCR	NR	2 days	Yes	DNS	2 days	NR	NR	NR	NR	NR	NR
5–7 day old Gn piglets	[61]	GII.4 (HS66)	oral	Yes; RT-PCR and ELISA	NR	4 days (range = 1–6 days)	Yes	DNS	4 days (range = 2–6 days)	NR	NR	Yes	Yes	IgM, IgA, and IgG	IgM, IgA, and IgG
2 day old Gn piglets	[63]	GII.4 (765)	oral	Yes; qRT-PCR	1 or 2 dpi	6 or 7 days	Yes	DNS	DNS	NR	Yes	Yes	NR	NR	NR
6–7 day old Gn piglets	[65]	GII.4 (HS194) [HuNoV only]	oral	Yes; qRT-PCR	1 dpi	11 days	NR	N/A	N/A	NR	NR	NR	NR	NR	NR
6–7 day old Gn piglets	[65]	GII.4 (HS194) [+ IFNα treatment]	oral	Yes; qRT-PCR	3 dpi	10 days	NR	N/A	N/A	NR	NR	NR	NR	NR	NR
11 or 13 day old Gn piglets	[65]	GII.4 (HS194) [HuNoV only]	oral	Yes; qRT-PCR	5 dpi	9 days	No	N/A	N/A	NR	No	Yes	NR	NR	NR
11 or 13 day old Gn piglets	[65]	GII.4 (HS194) [+ simvastatin treatment]	oral	Yes; qRT-PCR	2 dpi	15 days	No	N/A	N/A	NR	No	Yes	NR	NR	NR
6 day old Gn piglets	[67]	GII.4 (092895; 2006b variant)	oral	Yes; qRT-PCR	2 dpi	4 days	Yes	3 dpi	3 dpi	NR	NR	Yes	Yes	NR	NR
*RAG2*/*IL2RG* KO Gn piglets	[58]	GII.4 (092895; 2006b variant)	oral	Yes; qRT-PCR	1 dpi	max = 27 days	Yes	DNS	DNS	NR	NR	Yes	Yes	NR	NR
28 day old miniature piglets	[64]	GII.12/GII.3 (CAU140599)	oral	Yes; qRT-PCR	1 or 3 dpi (DNS)	1 day (DNS)	Yes (DNS)	1, 2, or 3 dpi (DNS)	DNS	NR	No	Yes	Yes	NR	NR
5 day old Gn calves	[74]	GII.4 (HS66)	oral	Yes; qRT-PCR and ELISA	1 dpi	3 days (range = 1–6 day(s))	Yes	2 dpi	3 days (range = 2–6 days)	N/A	Yes	Yes	Yes	IgM, IgA, and IgG	IgA and IgG
BALB/c mice	[44]	GII mix (GII.4 and GII.6)	oral + i.p.	NR; qRT-PCR	N/A	N/A	NR	N/A	N/A	N/A	No	NR	NR	NR	NR
*Rag*^−/−^*γc*^−/−^ BALB/c mice	[44]	GII mix (GII.4 and GII.6)	oral + i.p.	Yes; qRT-PCR	1 dpi	2 or 3 days	No	N/A	N/A	N/A	No	Yes	NR	NR	NR
*Rag*^−/−^*γc*^−/−^ BALB/c mice	[44]	GI + GII mix (GI.3, GII.4, and GII.6)	oral + i.p.	Yes; qRT-PCR	1 dpi	1 day	No	N/A	N/A	N/A	No	Yes	NR	NR	NR
*Rag*^−/−^*γc*^−/−^ BALB/c mice	[44]	GII mix (GII.4 and GII.6)	oral	Yes; qRT-PCR	1 dpi	1 day	NR	N/A	N/A	N/A	No	No	NR	NR	NR
*Rag*^−/−^*γc*^−/−^ BALB/c mice	[44]	GII mix (GII.4 and GII.6)	i.p.	No	N/A	N/A	NR	N/A	N/A	N/A	No	Yes	NR	NR	NR
Hum. *Rag*^−/−^*γc*^−/−^ BALB/c mice	[44]	GII mix (GII.4 and GII.6)	oral + i.p.	Yes; qRT-PCR	1 dpi	2 days	Yes	1 dpi	NR	N/A	No	Yes	NR	NR	NR
Hum. *Rag*^−/−^*γc*^−/−^ BALB/c mice	[44]	GI + GII mix (GI.3, GII.4, and GII.6)	oral + i.p.	Yes; qRT-PCR	1 dpi	1 day	No	N/A	N/A	N/A	No	Yes	NR	NR	NR
*Rag*^−/−^*γc*^−/−^ BL/6×BL/10 mice	[44]	GII mix (GII.4 and GII.6)	i.p.	No	N/A	N/A	NR	N/A	N/A	N/A	No	Yes	NR	NR	NR

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
