# Peer review of "Human Norovirus: Experimental Models of Infection"

_viruses, 2019, doi:10.3390/v11020151_

Round 1

Reviewer 1 Report

General Comments

This manuscript reviews animal models used for the study of human noroviruses (HuNoVs) and the development of helper virus-free reverse genetics systems for HuNoV. While the listing of animal models used is comprehensive, the experimental model of HuNoV replication in cell culture (permanent cell lines, B cells, enteroids of various host origin) is only marginally mentioned. The significance of the gut microbiome (bacteria) for HuNoV infection (a relatively recent research topic) should be highlighted. More attention should be given to the infectivity of HuNoVs in the natural host and in gnotobiotic piglets. The review on helper virus-free reverse genetics systems of HuNoVs should be expanded by pointing out still existing bottlenecks.

Specific Comments

           Line

11        Consider reading: … countermeasures. Experimental infection of human volunteers and use of closely related viruses such as murine NoV…

14        … that support HuNoV research. This review…

21        Ref. [1] is cited incompletely. Ref. [2] is very old; consider replacing it by:

Thorne LG, Goodfellow IG. Norovirus gene expression and replication. J Gen Virol. 2014 Feb;95(Pt 2):278-91.

Ref. [3] is obsolete and should be omitted.

26        Consider citation and summary of: Kroneman A, Vega E, Vennema H, Vinjé J, White PA, Hansman G, Green K, Martella V, Katayama K, Koopmans M. Proposal for a unified norovirus nomenclature and genotyping. Arch Virol. 2013 Oct;158(10):2059-68.

33        Consider citation of the low number of virus particles/ID50 of HuNoV in humans, which is much lower than that for gnotobiotic (gn) piglets as cited in ref [22].

Teunis PF, Moe CL, Liu P, Miller SE, Lindesmith L, Baric RS, Le Pendu J, Calderon RL. Norwalk virus: how infectious is it? J Med Virol. 2008 Aug;80(8):1468-76.

The following review is also important in the context:

Manuel CS, Moore MD, Jaykus LA. Predicting human norovirus infectivity - Recent advances and continued challenges. Food Microbiol. 2018 Dec;76:337-345.

See also:

Suñén E, Sobsey MD. Recovery and detection of enterovirus, hepatitis A virus and Norwalk virus in hardshell clams (Mercenaria mercenaria) by RT-PCR methods. J Virol Methods. 1999 Feb;77(2):179-87.

Straub TM, Höner zu Bentrup K, Orosz-Coghlan P, Dohnalkova A, Mayer BK, Bartholomew RA, Valdez CO, Bruckner-Lea CJ, Gerba CP, Abbaszadegan M, Nickerson CA. In vitro cell culture infectivity assay for human noroviruses. Emerg Infect Dis. 2007 Mar;13(3):396-403.

37        … [25]. Humans of all age groups are…

43        … or VP1 and VP2. The immunogenicity… [Avoid repetition to explain VP2.]

53        … clinical trials… Provide a reference.

57        … are unclear. Animal models used for HuNoV studies are summarized in Table 1 and have the potential to evaluate candidate HuNoV vaccines and therapeutics.

59        In vitro propagation of HuNoV. Here it should be noted that such systems are now available:

Jones MK, Watanabe M, Zhu S, Graves CL, Keyes LR, Grau KR, Gonzalez-Hernandez MB, Iovine NM, Wobus CE, Vinjé J, Tibbetts SA, Wallet SM, Karst SM. Enteric bacteria promote human and mouse norovirus infection of B cells. Science. 2014 Nov 7;346(6210):755-9.

Ettayebi K, Crawford SE, Murakami K, Broughman JR, Karandikar U, Tenge VR, Neill FH, Blutt SE, Zeng XL, Qu L, Kou B, Opekun AR, Burrin D, Graham DY, Ramani S, Atmar RL, Estes MK. Replication of human noroviruses in stem cell-derived human enteroids. Science. 2016 Sep 23;353(6306):1387-1393.

Failed attempts in permanent cell lines are reported in: Duizer E, Schwab KJ, Neill FH, Atmar RL, Koopmans MP, Estes MK. Laboratory efforts to cultivate noroviruses. J Gen Virol. 2004 Jan;85(Pt 1):79-87.

64        to 65. Consider omitting the last sentence.

66        Table 1. Animal models of HuNoV infection and disease. In the table spell out NHP. Line 3 of ref [40] can be omitted. In legend explain KO.

75        … antibody production, suggesting a lack of adaptation… [After a single passage this would not be expected.]

100      g.e. Spell out at first mentioning (genome equivalents). The ID50 of HuNoV for gn piglets is much higher than for humans (homologous natural transmission). This should be emphasized to signify a problem point of the gn piglet model for HuNoV transmission. See comments above.

116      Here the importance of the gut microbiome for viral infection (in particular HuNoV) should be emphasized. This kind of research is in its infancy but important in the context of this review. Consider citation of:

Pfeiffer JK, Virgin HW. Viral immunity. Transkingdom control of viral infection and immunity in the mammalian intestine. Science. 2016 Jan 15;351(6270). pii: aad5872.

Erickson AK, Jesudhasan PR, Mayer MJ, Narbad A, Winter SE, Pfeiffer JK. Bacteria Facilitate Enteric Virus Co-infection of Mammalian Cells and Promote Genetic Recombination. Cell Host Microbe. 2018 Jan 10;23(1):77-88.e5.

… and possibly expand.

135      In this context, the following citation is suggested:

Van Dycke J, Arnoldi F, Papa G, Vandepoele J, Burrone OR, Mastrangelo E, Tarantino D, Heylen E, Neyts J, Rocha-Pereira J. A Single Nucleoside Viral Polymerase Inhibitor Against Norovirus, Rotavirus, and Sapovirus-Induced Diarrhea. J Infect Dis. 2018 Jul 31. doi: 10.1093/infdis/jiy398. [Epub ahead of print]

142      … Vaccinia virus strain…

161      This part of review of ref [59] should be expanded.

167      Ref [14]. Apart from this being described as significant progress, the limitations and relative shortcomings of this system should be made more transparent, e.g. that the number of  HuNoV particles produced by helper virus-free reverse genetics was very low and that produced particles so far were not found to be infectious. The system was successful in producing infectious MuNoV particles.

181      The last sentence should be rephrased.

185      to line 204. It is very important that the Figure 1 is not split as shown, but reproduced undivided [best on p 8 of the       draft printout].

207      … needed to drive HuNoV pathogenesis studies and vaccine development.

211      … infection. Pathogenesis studies in response…

215      … NHPs are costly, and studies involving…

217      Ref [63] is outdated. Consider citation of:

Saif LJ, Ward LA, Yuan L, Rosen BI, To TL. The gnotobiotic piglet as a model for studies of disease pathogenesis and immunity to human rotaviruses. Arch Virol Suppl. 1996;12:153-61.

Yuan L, Ward LA, Rosen BI, To TL, Saif LJ. Systematic and intestinal antibody-secreting cell responses and correlates of protective immunity to human rotavirus in a gnotobiotic pig model of disease. J Virol. 1996 May;70(5):3075-83.

Saif L, Yuan L, Ward L, To T. Comparative studies of the pathogenesis, antibody immune responses, and homologous protection to porcine and human rotaviruses in gnotobiotic piglets. Adv Exp Med Biol. 1997;412:397-403.

224      … Also, mice lack the ability…

236      The Suppl Mat mentioned (Fig. S1, Table S1, Video S1) were not available for study.

242      Ref [1]: Green KY. Caliciviridae: The Noroviruses. In: Fields Virology, 6th ed (Knipe DM, Howley PM et al, eds), pp 582-608. Wolters Kluwer Health/ Lippincott Williams & Wilkins, Philadelphia PA, 2013.

Author Response

Response to Reviewer 1

We thank the reviewer for their helpful suggestions and respond to the comments accordingly:

General Comments

This manuscript reviews animal models used for the study of human noroviruses (HuNoVs) and the development of helper virus-free reverse genetics systems for HuNoV. While the listing of animal models used is comprehensive, the experimental model of HuNoV replication in cell culture (permanent cell lines, B cells, enteroids of various host origin) is only marginally mentioned.

·         The focus of animal models used to study human HuNoV was approved by the editors before we wrote the review, as we were informed that another manuscript in this issue was addressing HuNoV cultivation. We cited the B cell and enteroid systems as noted.  

The significance of the gut microbiome (bacteria) for HuNoV infection (a relatively recent research topic) should be highlighted.

·         We concur and have expanded the section on bacteria and enteric viruses as recommended on page 5 lines 278-280 and page 8 lines 445-449 in the revised manuscript.

More attention should be given to the infectivity of HuNoVs in the natural host and in gnotobiotic piglets. The review on helper virus-free reverse genetics systems of HuNoVs should be expanded by pointing out still existing bottlenecks.

·         The infectivity and promise of gnotobiotic pigs as animal models is noted on pages 4, 5, and 8 of the revised manuscript. We have discussed the inefficiency of current reverse genetic systems, the absence of clinically relevant systems, and some of the pitfalls in development of successful animal models and reverse genetic systems on page 6 lines 348-369 in the revised manuscript.

Specific Comments

Line:

11        Consider reading: … countermeasures. Experimental infection of human volunteers and use of closely related viruses such as murine NoV

·         Change accepted

14        … that support HuNoV research. This review… 

·         Change accepted

21        Ref. [1] is cited incompletely. Ref. [2] is very old; consider replacing it by:

·         We updated as requested. Reference 2 is the original observations regarding HuNoV biology. The review article below has been added as requested.

Thorne LG, Goodfellow IG. Norovirus gene expression and replication. J Gen Virol. 2014 Feb;95(Pt 2):278-91.

Ref. [3] is obsolete and should be omitted.

·         Reference 3 has been omitted.  

26           Consider citation and summary of: Kroneman A, Vega E, Vennema H, Vinjé J, White PA, Hansman G, Green K, Martella V, Katayama K, Koopmans M. Proposal for a unified norovirus nomenclature and genotyping. Arch Virol. 2013 Oct;158(10):2059-68.

·         Reference added and the classification of genogroups/genotypes is clarified to be based on VP1 amino acid homology is made on page 5 lines 5-8.

 33        Consider citation of the low number of virus particles/ID50 of HuNoV in humans, which is much lower than     that for gnotobiotic (gn) piglets as cited in ref [22]. Teunis PF, Moe CL, Liu P, Miller SE, Lindesmith L, Baric RS, Le Pendu J, Calderon RL. Norwalk virus: how infectious is it? J Med Virol. 2008 Aug;80(8):1468-76.

·         Reference added and contextualized with the references below showing a higher ID50 than previously thought.

Atmar, R. L. et al. Determination of the 50% human infectious dose for Norwalk virus. The Journal of infectious diseases 209, 1016-1022, doi:10.1093/infdis/jit620 (2014).

The following review is also important in the context: 

Manuel CS, Moore MD, Jaykus LA. Predicting human norovirus infectivity - Recent advances and continued challenges. Food Microbiol. 2018 Dec; 76:337-345. 

See also: 

Suñén E, Sobsey MD. Recovery and detection of enterovirus, hepatitis A virus and Norwalk virus in hardshell clams (Mercenaria mercenaria) by RT-PCR methods. J Virol Methods. 1999 Feb;77(2):179-87.

 Straub TM, Höner zu Bentrup K, Orosz-Coghlan P, Dohnalkova A, Mayer BK, Bartholomew RA, Valdez CO, Bruckner-Lea CJ, Gerba CP, Abbaszadegan M, Nickerson CA. In vitro cell culture infectivity assay for human noroviruses. Emerg Infect Dis. 2007 Mar;13(3):396-403.

37        … [25]. Humans of all age groups are… 

·         Change accepted

43        … or VP1 and VP2. The immunogenicity… [Avoid repetition to explain VP2.]

·         Change accepted

53        … clinical trials… Provide a reference.

·         References have been added. 

57        … are unclear. Animal models used for HuNoV studies are summarized in Table 1 and have the potential to evaluate candidate HuNoV vaccines and therapeutics.

·         Change accepted  

59        In vitro propagation of HuNoV. Here it should be noted that such systems are now available:

·         Text changed and HuNoV cell culture system references added.  

Jones MK, Watanabe M, Zhu S, Graves CL, Keyes LR, Grau KR, Gonzalez-Hernandez MB, Iovine NM, Wobus CE, Vinjé J, Tibbetts SA, Wallet SM, Karst SM. Enteric bacteria promote human and mouse norovirus infection of B cells. Science. 2014 Nov 7;346(6210):755-9. 

Ettayebi K, Crawford SE, Murakami K, Broughman JR, Karandikar U, Tenge VR, Neill FH, Blutt SE, Zeng XL, Qu L, Kou B, Opekun AR, Burrin D, Graham DY, Ramani S, Atmar RL, Estes MK. Replication of human noroviruses in stem cell-derived human enteroids. Science. 2016 Sep 23;353(6306):1387-1393.

Failed attempts in permanent cell lines are reported in: Duizer E, Schwab KJ, Neill FH, Atmar RL, Koopmans MP, Estes MK. Laboratory efforts to cultivate noroviruses. J Gen Virol. 2004 Jan;85(Pt 1):79-87.

64        to 65. Consider omitting the last sentence.

·         Sentence revised   

66        Table 1. Animal models of HuNoV infection and disease. In the table spell out NHP. Line 3 of ref [40] can be omitted. In legend explain KO.

·         Changes accepted; infection attempt of Cynomolgus macaque infection trial from ref [40] has been retained, but clarified, because this failed attempt at infection may inform future NHP HuNoV animal modeling/studies.  

75        … antibody production, suggesting a lack of adaptation… [After a single passage this would not be expected.]

·         The authors believe that the lack of effects following virus passage in vivo warrant inclusion, as other passage experiments have been performed (GN piglets), allowing for comparisons.

100      g.e. Spell out at first mentioning (genome equivalents).

·         Genome equivalents were first mentioned on line 82 where the acronym was established

The ID50 of HuNoV for gn piglets is much higher than for humans (homologous natural transmission). This should be emphasized to signify a problem point of the gn piglet model for HuNoV transmission. See comments above.

·         Agree; addressed above. 

116      Here the importance of the gut microbiome for viral infection (in particular HuNoV) should be emphasized. This kind of research is in its infancy but important in the context of this review. Consider citation of: 

Pfeiffer JK, Virgin HW. Viral immunity. Transkingdom control of viral infection and immunity in the mammalian intestine. Science. 2016 Jan 15;351(6270). pii: aad5872.

Erickson AK, Jesudhasan PR, Mayer MJ, Narbad A, Winter SE, Pfeiffer JK. Bacteria Facilitate Enteric Virus Co-infection of Mammalian Cells and Promote Genetic Recombination. Cell Host Microbe. 2018 Jan 10;23(1):77-88.e5.

… and possibly expand.

·         References added and the role of bacteria in enteric infections is emphasized.

135      In this context, the following citation is suggested: 

Van Dycke J, Arnoldi F, Papa G, Vandepoele J, Burrone OR, Mastrangelo E, Tarantino D, Heylen E, Neyts J, Rocha-Pereira J. A Single Nucleoside Viral Polymerase Inhibitor Against Norovirus, Rotavirus, and Sapovirus-Induced Diarrhea. J Infect Dis. 2018 Jul 31. doi: 10.1093/infdis/jiy398. [Epub ahead of print]

·         Viral polymerase inhibitor studies and reverse genetic systems were added

142      … Vaccinia virus strain…

·         Change accepted

161      This part of review of ref [59] should be expanded.

·         See above regarding viral polymerase inhibitors

167      Ref [14]. Apart from this being described as significant progress, the limitations and relative shortcomings of this system should be made more transparent, e.g. that the number of  HuNoV particles produced by helper virus-free reverse genetics was very low and that produced particles so far were not found to be infectious. The system was successful in producing infectious MuNoV particles.

·         Additional information regarding the outcomes of this system have been added. 

181      The last sentence should be rephrased.

·         Sentence rephrased

185      to line 204. It is very important that the Figure 1 is not split as shown, but reproduced undivided [best on p 8 of the       draft printout].

·         Figure 1 is contained in its entirety on page 7 of the revised manuscript.

207      … needed to drive HuNoV pathogenesis studies and vaccine development.

·         Change accepted

211      … infection. Pathogenesis studies in response… 

·         Phrasing altered

215      … NHPs are costly, and studies involving…

·         Change accepted

217      Ref [63] is outdated. Consider citation of: 

Saif LJ, Ward LA, Yuan L, Rosen BI, To TL. The gnotobiotic piglet as a model for studies of disease pathogenesis and immunity to human rotaviruses. Arch Virol Suppl. 1996; 12:153-61.

Yuan L, Ward LA, Rosen BI, To TL, Saif LJ. Systematic and intestinal antibody-secreting cell responses and correlates of protective immunity to human rotavirus in a gnotobiotic pig model of disease. J Virol. 1996 May;70(5):3075-83.

Saif L, Yuan L, Ward L, To T. Comparative studies of the pathogenesis, antibody immune responses, and homologous protection to porcine and human rotaviruses in gnotobiotic piglets. Adv Exp Med Biol. 1997; 412:397-403.

·         Reference added including: Yuan, L.; Jobst, P.M.; Weiss, M. Chapter 5 - Gnotobiotic Pigs: From Establishing Facility to Modeling Human Infectious Diseases. In Gnotobiotics, Schoeb, T.R., Eaton, K.A., Eds. Academic Press: 2017; https://doi.org/10.1016/B978-0-12-804561-9.00005-0pp. 349-368.

224      … Also, mice lack the ability…

·         Clarified in revised manuscript 

236      The Suppl Mat mentioned (Fig. S1, Table S1, Video S1) were not available for study.

·         There are no supplemental materials for this review. References to supplemental materials have been removed. 

242      Ref [1]: Green KY. Caliciviridae: The Noroviruses. In: Fields Virology, 6th ed (Knipe DM, Howley PM et al, eds), pp 582-608. Wolters Kluwer Health/ Lippincott Williams & Wilkins, Philadelphia PA, 2013.

·         Change accepted

Reviewer 2 Report

General comments:

This is a succinct, focused and comprehensive review of experimental animal models for HuNoV infection. Table 1 is comprehensive and helpful, but requires further edits to correct data and additions to provide HBGA profiles where  available in the papers or known (mice do not express human HBGA). Furthermore the influence of HBGA in susceptibility and binding to HuNoVs has been completely overlooked in this review, but based on the overwhelming evidence for their role in human susceptibility to NoV infection, they should be reviewed and discussed in regard to applicability and further development of the HuNoV animal models.

Also to provide additional perspectives, it would be useful to include a brief section on natural/experimental infection of animals with the host-specific NoVs and summarize differences and similarities to natural NoV infections in humans. This could include Murine NoV infections in wild type mice, bovine NoV infections and canine NoV infections, with the latter a potentially promising new animal model based on features very similar to HuNoV infections, including the role of HBGA binding.

This reviewer disagrees with the authors’ conclusions in lines 123-124 and the conclusions in lines 211- 213, that of all the large animal models, the GF calf best recapitulates HuNoV infection. Using both the GF calf and piglet animal models, investigators demonstrated mild diarrhea, virus shedding, viral antigen in enterocytes/mononuclear cells as well as seroconversion in both species and vaccine induced protection against homologous HuNoV challenge. Furthermore some HuNoV-infected GF pigs (Souza, et al J. Virol., 2007; Park et al 2018) and calves (Souza et al, 2008) showed small intestinal lesions/apoptotic cells and most developed NoV-specific IgA and IgG ASCs and antibodies in intestine and blood. On this basis both GF animal models appear promising and suitable for pursuing additional studies for testing of HuNoV vaccines and antivirals. However the association between HBGA type and susceptibility to HuNoV infection was only shown in the piglet model since pigs clearly express A+ and H+ HBGA on their enterocytes (and other cells as in humans) cross-reactive with those of humans (Cheetham, S., et al. 2007. Binding patterns of human Norovirus-like particles to buccal and intestinal tissues of gnotobiotic pigs in relation to A/H histo-blood group antigen expression. J. Virol. 81:3535-3544). This feature and the emetic response potential in pigs suggests the possibility of the greater applicability of the GF piglet model, although GF calves also express H+ related HBGA, but with undefined roles in HuNoV infection.

Specific Comments/suggested edits:

Abstract:

L12 change homologous to…use of related…

Text:

L 28 incorrect--GIV NoVs also infect dogs and cats/lions

L 46 should add information on chimps vaccinated IM with GI HuNoV VLPs and challenged with HuNoVs.

L48 add info on VLP induced protection

…and found to protect against homologous GII.4 HuNoV challenge.

…immunogenicity and protective efficacy of bovine NoV VLPs was tested in gnotobiotic calves and shown to provide partial protection against bovine NoV.

L57  …Development of translatable…

L58  …will help to evaluate

L61-62  …cell culture system is in progress to bypass…

L 84  … presentation and ethical considerations…

L110-111  Inaccurate as stated—the gnotobiotic pigs were not treated with simvastatin, but given IFN alpha alone—see ref 50 and also need to correct data in Table 1 to show groups +statin or no statin-- could also add IFN alpha groups, but need to define such groups

L123-124  Please modify to clarify. …immune responses. Revise as "Of all the large animal models, gnotobiotic calves infected with HuNoV, or more effectively and with more pronounced enteric disease with bovine NoV (Jena virus), better recapitulate the NoV enteric infection and disease evident in humans. An exception is that unlike pigs and humans, calves, like mice, do not have an emetic response that is often the dominant response to NoV infection in humans." The emetic aspect of NoV infection of humans and in animal models was largely overlooked, although its role in NoV pathogenesis is unclear.

L218-219 Although gnotobiotic animals lack commensals, this may be advantageous since they can readily be reconstituted with specific commensals or a human fecal microbiota  (Vlasova et al, mSphere 2:e00046-17, 2017) to remedy this concern. Also although germfree pigs and calves are immunocompetent at birth, they have lower immune responses than conventional animals. However after prior exposure to viruses or vaccination, enhanced and protective immune responses to enteric viral infections are induced that mimic those in conventional animals.

Table 1:

Corrections based on original data from ref 41,42, 50—authors should also verify data in table from other refs

Ref 41 L 6 Shedding and duration should be N/A since no infection

Ref 42 Duration 10-16 days, not 8 days

Ref 50 Indicate 2 groups as +satins –shedding onset 2dpi, duration 14 days

No statin –data in 2nd group OK but RNA detection info should be moved above for +statin group

Spelling errors- Abbreviations: intraperitoneal, gastro, lymph node

Author Response

Response to Reviewer 2

We thank the reviewer for their helpful suggestions and respond to the comments accordingly:

General Comments:

This is a succinct, focused and comprehensive review of experimental animal models for HuNoV infection. Table 1 is comprehensive and helpful, but requires further edits to correct data and additions to provide HBGA profiles where  available in the papers or known (mice do not express human HBGA). Furthermore the influence of HBGA in susceptibility and binding to HuNoVs has been completely overlooked in this review, but based on the overwhelming evidence for their role in human susceptibility to NoV infection, they should be reviewed and discussed in regard to applicability and further development of the HuNoV animal models.

·   This review centers on animal models used to study human HuNoV.  We appreciate the role of HBGAs in susceptibility and binding to HuNoVs, and mention the role for HBGAs as HuNoV attachment factors, but believe including a detailed description of these phenomena are outside the scope of this focused review. The rationale is that HBGAs are not the only HuNoV receptors (perhaps co-receptors) particularly as HBGA secretor-negative individuals can still become infected with HuNoVs (strong data in opposition to this is mostly associated with GI.1 Norwalk virus) and some HuNoVs display no HBGA binding capabilities.

Also to provide additional perspectives, it would be useful to include a brief section on natural/experimental infection of animals with the host-specific NoVs and summarize differences and similarities to natural NoV infections in humans. This could include Murine NoV infections in wild type mice, bovine NoV infections and canine NoV infections, with the latter a potentially promising new animal model based on features very similar to HuNoV infections, including the role of HBGA binding.

·   We agree that this information is important, but believe that the scope of natural NoV infections across all susceptible animal species is too large to be succinctly and adequately addressed in this focused review, and there are page limitations.

This reviewer disagrees with the authors’ conclusions in lines 123-124 and the conclusions in lines 211- 213, that of all the large animal models, the GF calf best recapitulates HuNoV infection. Using both the GF calf and piglet animal models, investigators demonstrated mild diarrhea, virus shedding, viral antigen in enterocytes/mononuclear cells as well as seroconversion in both species and vaccine induced protection against homologous HuNoV challenge. Furthermore some HuNoV-infected GF pigs (Souza, et al J. Virol., 2007; Park et al 2018) and calves (Souza et al, 2008) showed small intestinal lesions/apoptotic cells and most developed NoV-specific IgA and IgG ASCs and antibodies in intestine and blood. On this basis both GF animal models appear promising and suitable for pursuing additional studies for testing of HuNoV vaccines and antivirals. However the association between HBGA type and susceptibility to HuNoV infection was only shown in the piglet model since pigs clearly express A+ and H+ HBGA on their enterocytes (and other cells as in humans) cross-reactive with those of humans (Cheetham, S., et al. 2007. Binding patterns of human Norovirus-like particles to buccal and intestinal tissues of gnotobiotic pigs in relation to A/H histo-blood group antigen expression. J. Virol. 81:3535-3544). This feature and the emetic response potential in pigs suggests the possibility of the greater applicability of the GF piglet model, although GF calves also express H+ related HBGA, but with undefined roles in HuNoV infection.

·   We have modified the text on pages 4, 5, and 8 to better illustrate the potential of both the gnotobiotic pig and calf models. HGBA typing and analysis for HuNoV-infected GF piglets is mentioned as well as a lack of these studies for the other animal models. The emetic response potential for each animal model is discussed and the limitations for studying models that do not have an emetic response is mentioned.  

Specific Comments:

Abstract:

L12 change homologous to…use of related…

·   Change accepted 

Text:

L 28 incorrect--GIV NoVs also infect dogs and cats/lions

·   Text adjusted and cited accordingly. Specific studies are not cited because a number of these strains were classified as GIVs until the creation of new genogroups, which now contained those strains (namely canine GIVs à GVIs).

L 46 should add information on chimps vaccinated IM with GI HuNoV VLPs and challenged with HuNoVs.

·   Change accepted

L48 add info on VLP induced protection 

·   Change accepted

…and found to protect against homologous GII.4 HuNoV challenge.

·         Change accepted

…immunogenicity and protective efficacy of bovine NoV VLPs was tested in gnotobiotic calves and shown to provide partial protection against bovine NoV.

·         Change accepted

L57  …Development of translatable…

·         Text altered for clarity

L58  …will help to evaluate

·         Text altered for clarity

L61-62  …cell culture system is in progress to bypass…

·         Change accepted

L 84  … presentation and ethical considerations…

·         Added to the  Conclusions section

L110-111  Inaccurate as stated—the gnotobiotic pigs were not treated with simvastatin, but given IFN alpha alone—see ref 50

·         Change accepted and texted clarified

…and also need to correct data in Table 1 to show groups + statin or no statin-- could also add IFN alpha groups, but need to define such groups

·         + statin and + IFNa groups added to Table 1 and updated accordingly.

L123-124  Please modify to clarify. …immune responses. Revise as "Of all the large animal models, gnotobiotic calves infected with HuNoV, or more effectively and with more pronounced enteric disease with bovine NoV (Jena virus), better recapitulate the NoV enteric infection and disease evident in humans. An exception is that unlike pigs and humans, calves, like mice, do not have an emetic response that is often the dominant response to NoV infection in humans." The emetic aspect of NoV infection of humans and in animal models was largely overlooked, although its role in NoV pathogenesis is unclear.

·         Sentence modified; emetic response information has been added to the Conclusions section.

L218-219 Although gnotobiotic animals lack commensals, this may be advantageous since they can readily be reconstituted with specific commensals or a human fecal microbiota  (Vlasova et al, mSphere 2:e00046-17, 2017) to remedy this concern.

·         This is an important detail that was previously omitted. It is now included and the translatability of these bacterial studies to enhancement of HuNoV animal models has been emphasized.

Also although germfree pigs and calves are immunocompetent at birth, they have lower immune responses than conventional animals. However after prior exposure to viruses or vaccination, enhanced and protective immune responses to enteric viral infections are induced that mimic those in conventional animals. 

Table 1:

Corrections based on original data from ref 41,42, 50—authors should also verify data in table from other refs Ref 41 L 6 Shedding and duration should be N/A since no infection

·         Change accepted

Ref 42 Duration 10-16 days, not 8 days

·         Virus shedding does not begin until day 3 pi for these two piglets and ends at either day 10 or 16 pi resulting in a duration of 8 or 14 days.

Ref 50 Indicate 2 groups as +satins – shedding onset 2dpi, duration 14 days

No statin –data in 2nd group OK but RNA detection info should be moved above for +statin group

·         the two groups in Table 1 for reference 50 represent data from HuNoV only piglets from the IFNa experiments (6-7 day old GN piglets) , and HuNoV alone pigs from the simvastatin experiments (11 or 13 day old GN piglets). Now all 4 groups are represented in Table 1.

Spelling errors- Abbreviations: intraperitoneal, gastro, lymph node

·         Abbreviations fixed